# Specific Effects of Trabectedin and Lurbinectedin on Human Macrophage Function and Fate—Novel Insights

**DOI:** 10.3390/cancers12103060

**Published:** 2020-10-20

**Authors:** Adrián Povo-Retana, Marina Mojena, Adrian B. Stremtan, Victoria B. Fernández-García, Ana Gómez-Sáez, Cristina Nuevo-Tapioles, José M. Molina-Guijarro, José Avendaño-Ortiz, José M. Cuezva, Eduardo López-Collazo, Juan F. Martínez-Leal, Lisardo Boscá

**Affiliations:** 1Instituto de Investigaciones Biomédicas Alberto Sols (Centro Mixto CSIC-UAM), 28029 Madrid, Spain; apovo@iib.uam.es (A.P.-R.); marinamojena@iib.uam.es (M.M.); astremtan@iib.uam.es (A.B.S.); bvfernandez@iib.uam.es (V.B.F.-G.); anagomezsaez@gmail.com (A.G.-S.); 2Centro de Biología Molecular (Centro Mixto CSIC-UAM), Nicolás Cabrera S/N, Ciudad Universitaria de Cantoblanco, 28049 Madrid, Spain; cnuevo@cbm.csic.es (C.N.-T.); Jmcuezva@cbm.csic.es (J.M.C.); 3Centro de Investigación Biomédica en Red en Enfermedades Raras (CIBERER), 28029 Madrid, Spain; 4Pharma Mar SA, 28770 Colmenar Viejo, Spain; jose.guijarro@gmail.com (J.M.M.-G.); fmartinez@pharmamar.com (J.F.M.-L.); 5Instituto de Investigación Sanitaria La Paz (IdiPaz), Hospital Universitario La Paz, 28046 Madrid, Spain; joseavenort@gmail.com (J.A.-O.); elopezc@salud.madrid.org (E.L.-C.); 6Centro de Investigación Biomédica en Red en Enfermedades Cardiovasculares (CIBERCV), 28029 Madrid, Spain

**Keywords:** macrophage, trabectedin, lurbinectedin, tumor-associated macrophage, apoptosis, cell viability, caspases, respiration, calcium dynamics, mitochondria, ATP

## Abstract

**Simple Summary:**

Trabectedin and lurbinectedin are two potent onco-therapeutic drugs for the treatment of advanced soft tissue sarcomas. Here we show that, in addition to cancer cells, these molecules exert important effects on macrophages and tissue-associated macrophages altering metabolic functions, such as respiration and cell viability. Interestingly, the macrophages from one-fourth of healthy individuals exhibit apoptotic cell death when treated with these drugs at therapeutic doses. These data are relevant to understand the action of these molecules in the context of chemotherapy of oncologic patients.

**Abstract:**

Background: Tumor-associated macrophages (TAMs) play a crucial role in suppressing the immunosurveillance function of the immune system that prevents tumor growth. Indeed, macrophages can also be targeted by different chemotherapeutic agents improving the action over immune checkpoints to fight cancer. Here we describe the effect of trabectedin and lurbinectedin on human macrophage cell viability and function. Methods: Blood monocytes from healthy donors were differentiated into macrophages and exposed to different stimuli promoting functional polarization and differentiation into tumor-associated macrophages. Cells were challenged with the chemotherapeutic drugs and the effects on cell viability and function were analyzed. Results: Human macrophages exhibit at least two different profiles in response to these drugs. One-fourth of the blood donors assayed (164 individuals) were extremely sensitive to trabectedin and lurbinectedin, which promoted apoptotic cell death. Macrophages from other individuals retained viability but responded to the drugs increasing reactive oxygen production and showing a rapid intracellular calcium rise and a loss of mitochondrial oxygen consumption. Cell-membrane exposure of programmed-death ligand 1 (PD-L1) significantly decreased after treatment with therapeutic doses of these drugs, including changes in the gene expression profile of hypoxia-inducible factor 1 alpha (HIF-1α)-dependent genes, among other. Conclusions: The results provide evidence of additional onco-therapeutic actions for these drugs.

## 1. Introduction

Trabectedin (ET-743; TRB) is a tetrahydroisoquinoline compound that exhibits antitumor activity. Originally isolated form a tunicate that inhabits the Caribbean Sea, *Ecteinascidia turbinata*, this compound is currently produced synthetically and marketed as Yondelis^®^ (PharmaMar, Colmenar Viejo, Spain) [1]. It is structurally composed of three tetrahydroisoquinoline subunits [2] and acts as an intercalating DNA agent inducing an adduct formation in the DNA minor groove, causing torsion into the major groove [3]. It forms a Rad13–TRB–DNA ternary complex that breaks the double-stranded DNA helix and induces programmed-cell death [4].

TRB is indicated for the treatment of adult patients with advanced soft tissue sarcoma [5] after the failure of anthracyclines and ifosfamide or who are unsuited to receive these agents [6,7]. Trabectedin, in combination with pegylated liposomal doxorubicin, is indicated for the treatment of patients with relapsed platinum-sensitive ovarian cancer [6,7,8]. Moreover, it is used in non-operable liposarcomas and leiomyosarcomas [9,10], breast cancer with Breast Cancer type 1 Susceptibility Protein (BRCA1) loss [11], soft-tissue osteosarcoma and has been assessed in a phase II study of extraskeletal myxoid chondrosarcoma and a phase III study of mesenchymal chondrosarcoma [12]. Lurbinectedin (LUR) was approved by the Food and Drug Administration (FDA) in June 2020 after a phase II/III trial for the treatment of small cell lung carcinoma [13] and had previously been evaluated for ovarian cancer, breast cancer, sarcoma and acute myeloid leukemia (AML) [14,15,16,17].

The reported cytotoxic effects of these drugs at low nanomolar concentrations may not be sufficient to explain the severe monocytopenia and neutropenia suffered by cancer patients. This fact constitutes an exclusion criterion when dealing with these drugs [18]. Therefore, we hypothesized that there could be one or more underlying molecular mechanisms in addition to its DNA intercalation action targeting human immune cells.

Macrophages (MFs) are innate immune cells that display multiple and essential biological functions: these cells are considered phagocyte specialists. Thanks to their surface recognition pattern receptors, these cells can identify and bind pathogen-associated molecular patterns (PAMPs). These PAMPs bind to pattern-recognition receptors (PRRs), such as toll-like receptors (TLRs), which allow MFs to yield high antimicrobial capacity [19], their production of interferon-γ (IFNγ) explains its crucial role in neutralizing viral infections. These cells are distributed and located in practically all organs and tissues and, therefore, have a fundamental role in homeostasis and tissue regeneration by binding to damage-associated molecular patterns (DAMPs) that are exposed when viability or cellular function is compromised [20]. MFs orchestrate and regulate inflammatory processes by recruiting different cell types at the site of injury and are actively involved in subsequent tissue regeneration once inflammation is resolved [21]. They are considered a bridge between innate and adaptive immunity when they act as antigen presenting cells [22]. Its action is relevant in various inflammatory contexts such as cancer, where MFs could have a dual role: in the early tumor stages, MFs recognize and prevent the growth and development of tumor cells, which is an essential property of the immune surveillance, but if the neoplastic lesion somehow evades the immune system, the tumor microenvironment (tumor cells and stromal support cells) can recruit MFs and re-educate them to promote and maintain tumor progression. These cells suppress the antitumor action exerted by lymphoid cells in a physiological context [23]. This re-education of the phenotype by the MF is usually called tumor-associated macrophage (TAM).

Human MFs (hMFs) and dendritic cells (DCs) are considered the main contributors to side effects in various pathologies where inflammation and cancer are involved [24,25,26,27]. hMFs constitute a heterogeneous population of immune cells that adapt and respond rapidly to inflammatory signals. Generally speaking, two distinct subpopulations are defined among activated macrophages: classically activated toward a pro-inflammatory phenotype (upon stimulation with lipopolysaccharide (LPS), interleukin-1β (IL1β), IFNγ and tumor necrosis factor-α (TNFα); hM1, as acronym) and, alternatively, activated or anti-inflammatory macrophages (hM2, as acronym) [28]. hM1s mediate host defense responses against a wide range of pathogens and participate in anti-tumor immunity [24], while hM2s (induced by IL4, IL10, and IL13 exposure) mediate processes of homeostatic tissue remodeling and regulate wound healing, fibrosis, and tissue repair, once the initial inflammatory response reaches the resolution phase [29]. Furthermore, an additional functionally and metabolically intermediate phenotype has been described in the literature for the hTAM [23,30,31].

The polarization of hMFs is accompanied by drastic transcriptomic and metabolic reconfigurations [28,32]. Therefore, there is a current trend suggesting that re-education of hMFs toward a specific phenotype could be crucial to finding new therapeutic strategies in the context of tumor biology. However, this goal requires more extensive and deeper knowledge on the molecular basis of hMF polarization and DC activation [33]. Our data add new insights on the effects of TRB and LUR on the fate of hMFs, which may contribute to a better understanding of the antineoplastic actions of these drugs by altering the tumor microenvironment.

## 2. Results

### 2.1. Differential Responses of Human Monocyte-Derived Macrophages to Trabectedin and Lurbinectedin

Treatment of THP-1-differentiated macrophages with these drugs shows a significant loss in cell viability (with [I]_0.5app_ of 80 and 15 nM, respectively, at 24 h). A minimal rise in reactive oxygen species (ROS) production was observed at concentrations in the low-nM range (Figure 1a). When similar experiments were carried out in hMF, surprisingly ca. 77% of the donors (among 164 donors) were resistant to doses up to 100 nM (hMF-R), whereas 23% of the hMF from healthy donors exhibited a marked decrease in viability (hMF-S) (Figure 1b,c). A significant increase in ROS production was observed after treatment of hMF-R with TRB and LUR at high concentrations (Figure 1c), whereas this increase was only observed at concentrations of LUR <10 nM (Figure 1d). The dose-dependent curve on cell viability and ROS production of sensitive hMF (hMF-S) to TRB and LUR is shown in Figure 1d.

TRB- and LUR-resistant hMFs (hMF-R) exhibited similar distributions among the division cell cycle (Figure 2a), and viability was not affected by polarization to pro-inflammatory or anti-inflammatory cells (Figure 2b). Challenge with pro-inflammatory factors potentiates ROS production of hMF treated with TRB or LUR (Figure 2c). Interestingly, both drugs significantly decreased the phagocytic capacity of hMF-R cells (Appendix A).

### 2.2. Treatment of hMF with Trabectedin and Lurbinectedin Promotes Intracellular Ca^2+^ Rise and Changes in Mitochondrial Respiration

Incubation of hMF-R with TRB and LUR promotes a rapid rise in the cytoplasmic Ca^2+^ concentration (Figure 3a). The Ca^2+^ profiles are different, with a rapid increase in the case of TRB and a significant delay for LUR, probably reflecting the kinetics of the entrance of the drugs. This Ca^2+^ rise was reflected by phosphorylation of Acetyl-CoA Carboxylase (ACC) at 30 min, probably a time at which TRB-dependent calcium changes were abolished (Figure 3b). Moreover, when the mitochondrial membrane potential was measured, treatment of hMF-R with TRB or LUR showed hyperpolarization (Figure 3c) together with an increase in the cellular levels of ATP (Figure 3d). These results suggested that mitochondrial function was affected by the drugs. Measurement of the oxygen consumption rate of THP-1, hMF-S, and hMF-R cells treated with TRB or LUR showed different patterns of behavior (Figure 4). THP-1 and hMF-S cells showed a depressed respiratory capacity upon treatment with the drugs. However, hMF-R cells retained a significant basal respiratory capacity when compared with the drug-sensitive counterparts at 5 nM. These data point to mitochondrial function as one of the additional targets for these drugs in hMFs. Moreover, previous studies in tumor cells showed that TRB promotes a time-dependent dephosphorylation of the repeated heptapeptide Tyr–Ser–Pro–Thr–Ser–Pro–Ser of RNA pol II [34,35,36]. Analysis of the effect of TRB on the phosphorylation state of RNA pol II from hMF-S and hMF-R showed similar time courses of dephosphorylation (from the hyperphosphorylated to the hypo/de-phosphorylated state; IIo to IIa bands), which suggests a similar mechanism of action of the drug regardless hMF S or R typification (Appendix A).

### 2.3. Glutamine Metabolism Is Required for Survival of hMF-R Treated with Trabectedin and Lurbinectedin

To gain insight on the metabolic requirements of hMFs to retain viability upon treatment with TRB or LUR, cells were incubated with inhibitors of glycolysis (3PO), fatty acid synthesis (TOFA), or glutamine metabolism (C968). As Figure 5a shows, the inhibition of glutamine metabolism renders the cells to a TRB- and LUR-sensitive phenotype, probably unrelated to the observed increase in ROS production. Incubation of hMF-R cells with several metabolites that can influence the fate of hMFs (lactate, as final product of glycolysis; succinate or fumarate), failed to affect hMF-R viability, despite significantly increasing ROS production (Figure 5b).

In order to analyze the impact of different pathways involved in hMF function, a panel of activators/inhibitors were tested on the viability of hMF-S and hMF-R cells. As Figure 6 shows, only dexamethasone and IFNα treatment significantly enhanced hMF-S cell survival, whereas none of the tested molecules in hMF-R cells decreased their viability upon TRB or LUR challenge.

### 2.4. Specific Gene Profiling of hMF-R Treated with Trabectedin and Lurbinectedin

Because TRB and LUR seem to have specific effects on hMFs we proceeded to investigate the expression profile of a series of genes that may be relevant to understanding the impact of these drugs on hMF physiology. As Figure 7 shows, the effect of TRB and LUR on gene expression is dependent on the polarization phenotype of macrophages. We have investigated the response of representative genes in cells treated with LPS + IFNγ + IL1β (as pro-inflammatory hMFs), IL4 + IL10 + IL13 (as representative of anti-inflammatory/pro-resolution cells) or hMFs co-incubated with MDA cells (hTAM phenotype). The results show a significant impact of LUR on *HIF1A* upregulation that has a counterpart response in downstream regulated genes (e.g., *PFKFB3*, *CD274*, and *PDCD1LG2*), and an upregulation of *GPR132* by TRB. In addition to this, genes encoding for scavenger receptors, such as CD36 and Scavenger Receptor Class B Type 1 (SR-BI) exhibited a significant downregulation in hMFs treated with TRB or LUR regardless of the polarization condition of the hMFs. Together, these data suggest the potential unique and specific actions of each drug.

Interestingly, at the protein level, surface programmed-death ligand 1 (PD-L1) levels significantly decrease in hMF-R treated with TRB and LUR (Figure 8a). Moreover, the decrease of viability of hMF-S after TRB and LUR, is also observed in hTAM-S (Figure 8b), this may contribute to enhance the immune function over tumor cells. Regarding the mechanism by which TRB and LUR decrease the cell viability of hMF-S, we observed a dose-dependent accumulation of annexin V-positive cells, a situation that was not observed in the hMF-R counterparts (Figure 8c). Although hMF-R and hTAM-R cells exhibit a dose-dependent increase in ROS production (Appendix A).

Analysis of the mechanism involved in the cell death of hMF-S treated with TRB showed a time-dependent decrease in B-cell lymphoma 2 (Bcl-2) levels, an increase in the accumulation of the caspase 3 cleaved protein, and an increase in active poly (ADP)-ribose polymerase 1 (PARP1) (Figure 9a). Labeling of cells with specific targets of caspase 8 and caspase 9 revealed a prominent increase in caspase 9-positive cells in hMF-S, but not in the hMF-R counterparts, indicating the occurrence of a classic mitochondrial signaling-dependent caspase activation that was blocked by the pan-caspase inhibitor z-VAD (Figure 9b).

## 3. Discussion

Macrophages have been considered as key components for the survival of many tumors [31,37,38,39]. These tumor-associated macrophages (hTAMs) are educated by cancer cells and the stromal microenvironment to block the action of the adaptive immune system to fight and prevent tumor growth [38,40,41,42,43]. Not only do hTAMs contribute to avoiding immunosuppression, but they also participate in other tumor-supporting functions, such as neovascularization and the spread of tumor cells. Several previous works have analyzed the role of TRB and LUR in macrophage function [31,38,40,44,45,46,47]; however, after analyzing hMFs from monocyte-derived blood cells from nearly 200 different donors, we were surprised by the fact that ca. a quarter of hMFs were extremely sensitive to these drugs at the therapeutic doses used. In our opinion, this is important to understand the mechanism of action of these drugs and to develop stratified approaches for their therapeutic use. Indeed, both hMFs and hTAMs derived from these monocytic blood cells exhibit the same extreme sensitivity, and this response is consistently observed after blood donation at different times. Furthermore, complete depletion of hMF-S is observed in cell culture after treatment at doses in the 50–100 nM range of TRB or LUR, indicating that this is a specific cellular response for all hMF-S cells present in primary cultures. Opposite to this behavior, ca. 75% of the donors assayed were generating hMF-R and hTAM-R cells, very resistant to both drugs. Occasionally, we had the opportunity to evaluate samples from the same resistant donors at different times and, interestingly, the nature of the response (resistance) prevailed over time. In this work, we have used activators and inhibitors of many signaling pathways attempting to reverse the sensitive to the resistant phenotype and vice versa, to obtain sensitive cells from resistant donors. Only treatment with dexamethasone and IFNα could increase the resistance of hMF-S cells. However, activators of other nuclear receptors (e.g., peroxisome proliferator-activated receptor-γ (PPARγ), liver X receptor-α (LXRα)) were unable to mimic the action of the corticosteroid. In addition to this, inhibition of macrophage metabolic pathways revealed a key dependence on Gln metabolism since inhibition of glutaminase with C968 in cells treated with TRB or LUR dramatically reduced cell viability; however, inhibition of “forced glycolysis” with 3PO (a 6-phosphofructo-2-kinase/fructose-2,6-bisphosphatase 3 (PFKFB3) inhibitor) or fatty acid synthesis by TOFA (an ACC inhibitor) did not affect cell viability [28,48,49,50,51,52].

From a metabolic point of view, it has been suggested that intense aerobic glycolysis of tumor cells, which leads to large accumulations of lactate in the medium, contributes to retaining a pro-tumor phenotype of TAMs. However, we did not observe any effect of lactate on cell viability or ROS production from hMFs treated with TRB or LUR. Furthermore, other molecules that have been involved in MF polarization, such as extracellular succinate (via GPR91) or dimethyl fumarate did not influence the response to the drugs. This study was based on the observation that TRB increased *GPR91* levels in hMFs. Only the polarization of hMFs with proinflammatory factors and, to a lesser extent, with anti-inflammatory stimuli, improved ROS production without affecting cell viability after drug treatment. This is important to disclose any potential action of TRB and LUR increasing ROS production as part of the therapeutic effect of these drugs. In fact, treatment with TRB and LUR enhances ROS production by hMFs, and this had no effect on cell viability, at least during the 24–48 h in which we measured this enhanced ROS synthesis. In addition, these observations guaranteed the fact that TRB and LUR easily access hMF-R cells.

Analysis of the gene expression patterns of hMF-R cells differentiated into pro-inflammatory (M1), anti-inflammatory/pro-resolution (M2) or hTAMs showed complex and specific patterns of gene expression. A tendency to increase *HIF1A* was observed in response to LUR. This effect was more evident in both M1 and M2 hMF-R cells and was paralleled by the expression of HIF-1α transcription-dependent genes, such as *PFKFB3* and the genes encoding for PD-L1 and PD-L2. The ability of TRB to interfere in the transcription machinery had been previously described [2,8,46,53]. However, regardless of the hMF-S or hMF-R phenotype, all hMFs exhibited a time-dependent dephosphorylation of RNA polymerase II, suggesting a rapid action of TRB at the transcription level, but also that this effect is per se unable to promote hMF cell death [36].

Regarding drug-activated signaling pathways, independently of the drug-resistant or -sensitive phenotype of hMFs, a rapid and transient increase in intracellular Ca^2+^ levels was observed, but they exhibited specific profiles. While TRB promoted an immediate and transient dose-dependent Ca^2+^ elevation, the action of LUR exhibited a significantly delayed Ca^2+^ response. Although the mechanisms of internalization of both drugs in macrophages remain unclear [54,55,56,57], this difference in Ca^2+^ mobilization probably reflects unusual mechanisms in the way in which drugs are internalized to exert their action and could explain the pharmacodynamic differences between them. Interestingly, both drugs differentially affected mitochondrial membrane potential and respiratory capacity. hMF-R cells exhibited a significant hyperpolarization of the ΔΨm value [58], which is absent in drug-challenged hMF-S mitochondria at therapeutic concentrations and that has been described to preserve macrophage cell function [58]. In addition to this, the basal respiratory capacity of hMF-S cells was depressed after treatment with therapeutic doses of TRB and LUR. Indeed, the impact of TRB and LUR on the mitochondrial respiration was one of the clearest effects of these compounds on hMF physiology, which had not been described before. Interestingly, intracellular ATP levels exhibit a time-dependent elevation in cells treated with TRB and LUR, suggesting that cell death is not associated with a loss of ATP levels.

We focused our interest in understanding how TRB and LUR promote death in hMF-S cells. Analysis over time of annexin V exposure and propidium iodide permeability suggested that caspases were involved in this process. In fact, a time-dependent decrease in Bcl-2 levels was observed by immunoblotting. This was interesting since the half-life of Bcl-2 in macrophages is ca. 10–20 h [59], suggesting an active degradation of the protein upon drug treatment. Cleavage of caspase 3 due to the activation of caspase 9 was evident in these assays. In parallel to caspase 3 activation, cleavage of PARP-1 was observed, suggesting a functional role for this pro-apoptotic process in the decrease of hMF-S cell viability. This activation of PARP-1 has been described as one of the therapeutic actions against different tumor cells [6,11]. In fact, depletion of hTAMs by LUR has been described as one of the mechanisms of action against pancreatic ductal adenocarcinoma [47].

One additional observation was the decrease in the cell surface levels of PD-L1, which might contribute to reinforce the immune surveillance action of the adaptive immune system against tumor cells [43,60].

Taken together, our data unravel specific mechanisms of action of TRB and LUR on hMFs that can be classified as sensitive or resistant to the drug. It remains unclear whether the evaluation of this hMF phenotype could influence the outcome of pharmacological treatment. Since the generation of hMFs by circulating monocytes is a rapid and well-established protocol, further analysis of the therapeutic action of TRB and LUR under the stratification criteria of sensitive/resistant hMFs may contribute to establishing more personalized protocols for a better success in specific tumors treatments.

## 4. Materials and Methods

### 4.1. PharmaMar Antitumoral Drugs

Trabectedin (Yondelis^®^) and lurbinectedin (PM01183) were prepared in low nanomolar concentrations (0–200 nM). Final assay dilutions were prepared in RPMI 1640 + 2% fetal bovine serum (FBS) media, which were obtained from intermediate dilutions, 10 µM in DMSO for each compound whose original stock dilutions were 1 mM (DMSO). Stock solutions were periodically provided by PharmaMar (Colmenar Viejo, Spain).

### 4.2. Human Samples

Human buffy coats from healthy anonymous donors were obtained and isolated at IdiPAZ (Madrid). All the participants provided written consent in accordance with the ethical guidelines of the 1975 Declaration of Helsinki and the Committee for Human Subjects of La Paz University Hospital (HULP: PI-3521). Fresh blood was collected in anticoagulant EDTA-treated tubes for further fractioning into serum and peripheral blood mononuclear cells (PBMCs).

### 4.3. Human PMBC Isolation

Human peripheral blood mononuclear cells (PMBCs) were isolated from buffy coats by a Ficoll (Sigma, GE 17-0300, Madrid, Spain) gradient centrifugation in 50 mL following a previous protocol [61,62], in Falcon tubes, by gentle dripping of the buffy coat over 13 mL of Ficoll up to 50 mL (Ficoll + buffy coat) with a 25 mL sterile pipette, avoiding the mixture of both components, preserving the two different phases. It was subsequently spun at 652 *g* for 25 min at room temperature (22–24 °C) without break to prevent gradient loss. Plasma and PMBC fraction were retrieved from the upper-aqueous phase with a plastic, sterile pipette and recollected in another 50 mL Falcon tube. Then, to remove Ficoll traces, PMBCs were washed two times with sterile PBS (Lonza, BE17-515F, Madrid Spain) by serial centrifugations at 300 *g* for 5 min. Lysis buffer (Stem Cell catalog ref #07850) was added for lysing the remnant erythrocytes, followed by washing with PBS as mentioned before. A 100 µL aliquot was retrieved and analyzed by flow cytometry (FACS-Canto II, Becton Dickinson 338962) to determine the number of monocytes, and they were additionally counted by trypan blue exclusion (Sigma, T8154).

### 4.4. hMF Culture and Differentiation

Human monocytes were resuspended in serum-free DMEM (GIBCO, 11966-025, Madrid, Spain) plus penicillin/streptomycin (P/S 1%; GIBCO 15140/122) and cultured for 1 h at 37 °C + 5% CO_2_, to induce monocyte cell adhesion to 6 MW cell plates (Falcon, 353046, Fisher Scientific, Madrid, Spain), 12 MW plates (Falcon, 353043), or P100 plates (Falcon, 353003), respectively. Cells were washed two times with sterile PBS to remove lymphocytes, and cell media were replaced with DMEM + 10% FBS (GIBCO 10270/106). Cells were incubated for 10–14 days, allowing human monocytes (hMo) differentiation into hMFs [61]. Cell culture medium was renewed once a week. Once these cells were fully differentiated, experiments were carried out in RPMI 1640 (GIBCO 21875) +2% FBS. CD14^+^-cells were >90%.

### 4.5. THP-1 Culture and Differentiation

THP-1 (ATCC, TIB-202, Barcelona, Spain) is a human tumor cell line derived from acute monocytic leukemia. It was cultured in RPMI 1640 + P/S + 10% FBS in suspension. In order to differentiate THP-1 cells, 100 nM phorbol myristate acetate (PMA) (Sigma P1585) was added to the culture medium for 24 h, inducing cell adhesion. Cells were washed with sterile PBS, cultured for 24 h with RPMI 1640 + 10% FBS. Once these cells were fully differentiated, experiments were carried out in RPMI 1640 (GIBCO 21875) + 2% FBS.

### 4.6. hMF Polarization Assays

hM1 polarization [28] was carried out by incubating for 24 h hMFs with the following bioactive molecules: LPS (0.5 µg/mL; LPS-EB Ultrapure, InvivoGen, 5 × 10^6^ EU, Cat#tlrl-3pelps, Ibian Technologies, Zaragoza, Spain), human recombinant IL1β (PeproTech, 200-01B; 20 ng/mL, London, UK) and recombinant IFNγ (PeproTech, 300-02; 20 ng/mL), and TNFα (PeproTech, 300-01A; 20 ng/mL), followed by challenging with either TRB or LUR. hM2 polarization was carried out by incubating for 24 h hMFs with the following combination of human recombinant cytokines: IL4 (PeproTech, 200-04; 20 ng/mL), IL10 (PeproTech, 200-10; 20 ng/mL), and IL13 (PeproTech, 200-13; 20 ng/mL).

### 4.7. MDA-MB-231 Culture and hTAM Generation

MDA-MB-231 (ATCC, HTB-26, Barcelona, Spain) is a human tumor cell line of adenocarcinoma derived from a metastatic site (pleural effusion). These cells were cultured in RPMI 1640 + P/S + 10% FBS and used for hTAM generation. Once hMFs were fully differentiated in a 12 MW (1 × 10^6^ cells/well) a Transwell (Costar, 3460) where MDA-MB-231 was previously seeded (125,000 cells/well) was placed above the hMFs. Pore size (0.4 µm) exclusively allowed soluble metabolites to contact hMFs. The co-culture was maintained for 24 h. This conditioned media were preserved for the subsequent experiments.

### 4.8. Flow Cytometry Assays

Flow cytometry experiments were carried out in a FACS-Canto II (Becton Dickinson, 338962, Madrid, Spain). Differentiated hMFs and THP-1 supernatants were preserved, cells were trypsinized for 4 min at 37 °C + 5% CO_2_, and trypsin was neutralized with sterile PBS + 2% FBS. Cells were gently scrapped and centrifuged at 300 *g*, at room temperature for 5 min. Cells were then incubated with different fluorochromes for 30 min (unless indicated otherwise). Cell media supernatants were always centrifuged and properly considered for all cell viability determinations. Cell viability was determined by DAPI staining (2 µM, Life Technologies, Madrid, Spain) and incubating for 5 min at room temperature [28,32].

ROS production was measured by incubating cells for 30 min at 37 °C + 5% CO_2_ in darkness with 5 µM DCFH-DA fluorescent probe (2′-7′-dichlorofluorescein diacetate, D6683 Sigma).

Cell cycle experiments were assessed by 20 ng/mL PI staining (Sigma, 81845) for an incubation period of 15 min at room temperature in darkness. It was occasionally used to determine cell viability, in that case, PI concentration was 4 ng/mL, and the incubation period was 5 min.

Mitochondrial membrane potential (ΔΨm) measures in hMFs were monitored by 100 nM CMXROS (Red MitoTracker, Invitrogen, M7512). The fluorescent probe was incubated for 30 min at 37 °C + 5% CO_2_ in darkness [58].

PD-L1 measurements in hMFs and hTAMs were performed by using a primary polyclonal PD-L1 rabbit anti-human antibody (Abcam ab.209959, Madrid, Spain). Briefly, 5 × 10^5^ cells were incubated with 100 µL/sample of 1 µg/mL of the antibody for 30 min at 37 °C + 5% CO_2_ in darkness. The labeling excess was removed by washing twice with sterile PBS1X + 2% FBS (300 *g*, room temperature, 5 min). Then, it was incubated with a secondary goat anti-rabbit antibody (Molecular Probes, A11055). Sample were washed as described for the primary antibody and, finally, incubated with DAPI for 5 min prior to flow cytometry.

### 4.9. Immuno-Metabolic Assays

Inhibitors and activators were added 2 h prior to the treatment with the antitumoral agents (24 h). 3PO (Sigma, SML-1343; 10 µM), TOFA (Sigma, T6575; 1 µg/mL), C968 (Calbiochem, 352010; 40 nM, Madrid, Spain), sodium lactate (Sigma, L7022), sodium succinate dibasic hexahydrate (Sigma, S9637), di-methyl-fumarate (Sigma, 242926); lactacystine (Sigma, L6785; 10 µM), BBG (BioRad 161-0400; 10 µM, Alcobendas, Spain), LY294002 (Calbiochem, 440202; 10 µM), NFκB inhibitor SN50 (Calbiochem, 481480; 18 µM), actinomycin D (GE HealthCare Biosciences, 116128; 3 µg/mL), insulin (Sigma, 91077C; 100 nM), IFNγ (PeproTech, 300-02; 20 ng/mL), dexamethasone (Sigma, D4902; 1 µg/mL), rapamycin (Sigma, R0395, 10 nM), cycloheximide (Sigma, C6255; 10 µg/mL), rosiglitazone (Calbiochem, CAS155141-29-0; 1 µM), IFNα (Sigma, IF007, 20 ng/mL), MCC950 (Sigma, PZ0280; 300 nM), GW3965 (Merck, G6295, 1 μM), RAF inhibitor (Sigma, 475958; 10 μM), and PD98059 (Merck, HY-12028; 10 μM)

### 4.10. Apoptosis Assays

Annexin V measurements were done following the manufacturer instructions: Cells from the supernatant (dead cells) were centrifuged and resuspended in Buffer 1x (Immunostep, 556454, Salamanca, Spain) Annexin V (Immunostep, 556454) was added at 5 µL/sample, and incubated for 15 min at 37 °C + 5% CO_2_ in darkness; DAPI was subsequently added. BioVision kits were used for caspase 8 (#K188-25; Madrid, Spain) and caspase 9 (#K189-25) in vivo activity measurements [58,63]. As negative controls, z-VAD-FMK was added for every experimental condition, following the instructions of the supplier.

### 4.11. hMF Phagocytosis

A phagocytosis kit was used (BioParticles) of zymosan conjugated with Alexa Fluor 488 nm fluorescent probe (Invitrogen, Z23373, Madrid, Spain). In this case, cells were exposed to this specific fluorescent FITC-labeled compound for 2 h at 37 °C + 5% CO_2_ in darkness. Cells were washed twice with sterile PBS and then were trypsinized and scrapped off the plate as described above.

### 4.12. Cell Observer Measurements (Calcium Time-Lapse Assays)

Calcium measurements were carried out using a Fluo-4 Direct Calcium Assay Kit (Invitrogen, F10471) following the manufacturer’s instructions. Thapsigargin (Sigma, T9033; 200 nM) was used to determine the maximal calcium mobilization. Seven frames per second were registered. Experiments were recorded for 10 min each, in triplicate. At least 200 cells per donor and per experimental condition were monitored. In vivo cell observer (Carl Zeiss, Tres Cantos, Spain) is composed by an inverted, light-transmitted Z1 Observer microscope coupled with a tightly regulated cell incubator chamber (37 °C + 5% CO_2_) connected to a monochrome, high-resolution camera (Cascade 1K), 10x Plan-APOCHROMAT camera lens. Axiovision-4.8 Software Analysis as well as Image J were applied for the analysis of time-lapse Ca^2+^ dynamics.

### 4.13. ATP Measurements in hMFs

Cells were washed three times with ice-cold sterile PBS, trypsinized, pelleted, and treated with 5% trichloroacetic acid, cells were vortexed and neutralized with 3M KOH. ATP measurements were carried out using a bioluminescence ATP Determination Kit (Biaffin, GmbH and CoKG, Kassel, Germany) following the manufacturer’s instructions. ATP measurements were done in triplicate and internal standards of ATP were used for quantification.

### 4.14. Protein Analysis and Immunoblotting

Cells were homogenized at 4 °C in a lysis buffer containing 10 mM Tris-HCl, pH 7.5; 1 mM MgCl_2_, 1 mM EGTA, 10% glycerol, 0.5% CHAPS (Sigma, C3023), and protease and phosphatase inhibitor cocktails (Sigma, P8340, P5726, P0044). Samples were gently vortexed for 45 min and supernatants were spun at 12,000 rpms for 15 min at 4 °C. Total protein extracts (supernatants) were retrieved and stored at −20 °C. Total protein concentrations were determined by Bradford’s method.

Protein extracts were prepared with sample buffer (60 mM Tris.HCl, pH 6.8, 2% SDS, 40% glycerol, 3% β-mercaptoethanol, 0.005% blue bromophenol) and were subsequently boiled at 95 °C for 5 min in a thermo-block. Next, 40 µg of total protein was loaded per well in SDS-PAGE following previous protocols [28]. Membranes were incubated O/N (4 °C) with the following primary antibodies and for 1 h at room temperature (RT) with the secondary antibodies, washing for 4 times (Table 1). Uncropped Western Blot Images can be found at Appendix A.

### 4.15. Seahorse (Agilent Technologies XF24)

Oxygen consumption rate (OCR) was measured in real time, following the instructions of the manufacturer (Agilent, 103576-100, Las Rozas de Madrid, Spain). Seahorse analyser was calibrated with a calibrating Seahorse XF solution (Agilent, 103059-000). Respiratory chain inhibitors were used at these concentrations: 6 μM oligomycin, 0.75 mM DNP (2′,4′-dinitrophenol), 1 μM rotenone, and 1 μM antimycin A [64]. 

### 4.16. RNA Isolation and Analysis

Cells were washed with ice-cold sterile PBS. Trizol^®^ Ambion (Madrid, Spain) protocol was applied for RNA extraction (1 mL/plate sample) [28]. RNA was quantified by ND-1000 (Nanodrop Technologies; Madrid, Spain) and the RNA integrity number (RIN) was calculated. RNA was retrotranscripted to cDNA with Transcriptor First-Strand DNA cDNA Synthesis (Roche) following the manufacturer’s instructions. qPCR assays were carried out with 5 µL cDNA + 10 µL SYBR^®^ Green PCR Master Mix cocktail (Applied Biosystems) + 250 nM forward and reverse primers (Table 2). *RLPLP0* was chosen as a house-keeping endogenous control for normalization purposes. Three RNA pools of four donors each were evaluated for each condition, by triplicate. qPCR reaction was carried out in MyIQ RealTime PCR System (BioRad). Result analysis was conducted with IQ5 program (BioRad), following the ΔΔCt method.

### 4.17. Statistical Analysis

All results are depicted as mean ± S.E.M. Two-tailed paired Student-T tests were performed. Statistical significance: * *p* < 0.05, ** *p* < 0.01, *** *p* < 0.001. Data were analyzed by the SPSS for Windows statistical package, v21 and GraphPad Prism 8.0.0. (GraphPad Software, San Diego, CA, USA.)

## 5. Conclusions

The data reported in this work, using human macrophages derived from circulating monocytes, show the existence of two different behaviors in response to TRB and LUR. This diversity is important to understand the outcomes of the immune surveillance over tumor cells. Additionally, the data reported in this work point to the mitochondria as key players in the initiation of human macrophage apoptotic death in TRB- and LUR-sensitive cells. Although the molecular mechanism(s) that support(s) these differential responses remain yet unidentified, assessment of this condition in the therapeutic action of the drugs may contribute to design new intervention protocols to fight cancer.

## Figures and Tables

**Figure 1 cancers-12-03060-f001:**
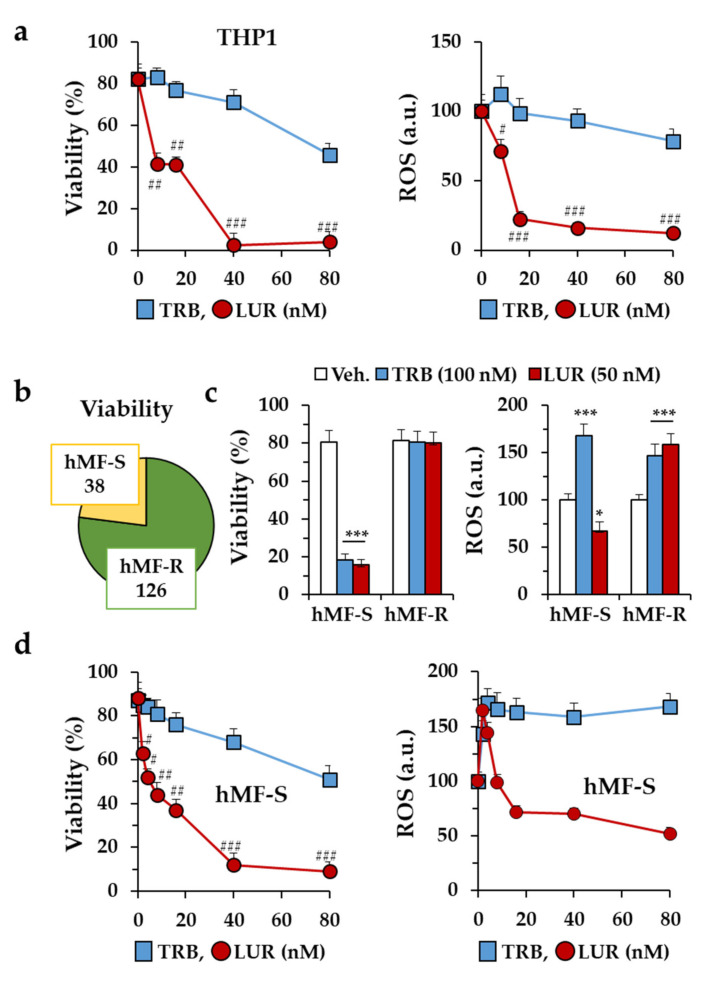
Effect of trabectedin (TRB) and lurbinectedin (LUR) on the viability of human macrophages. (**a**) Dose-dependent effect on cell viability and reactive oxygen species (ROS) production of TRB and LUR on THP-1 macrophages after 24 h of incubation; (**b**) distribution of TRB- and LUR-sensitive (hMF-S) and -resistant (hMF-R) human macrophages (hMFs) differentiated from monocytes from healthy donors; (**c**) effect on cell viability and ROS production of hMFs after 24 h of treatment with 100 nM TRB and 50 nM LUR; (**d**) dose-dependent effect of TRB and LUR on cell viability and ROS production of hMF-S. Results show the means + S.E.M. of 4 independent preparations of THP-1 cells, or the means + S.E.M. of cells from 8 (panels a,d) and 12 (panel c) different donors. * *p* < 0.05; *** *p* < 0.005 vs. the vehicle (Veh.) condition; ^#^
*p* < 0.05; ^##^
*p* < 0.01; ^###^
*p* < 0.005 vs. the corresponding condition with TRB. a.u., arbitrary units.

**Figure 2 cancers-12-03060-f002:**
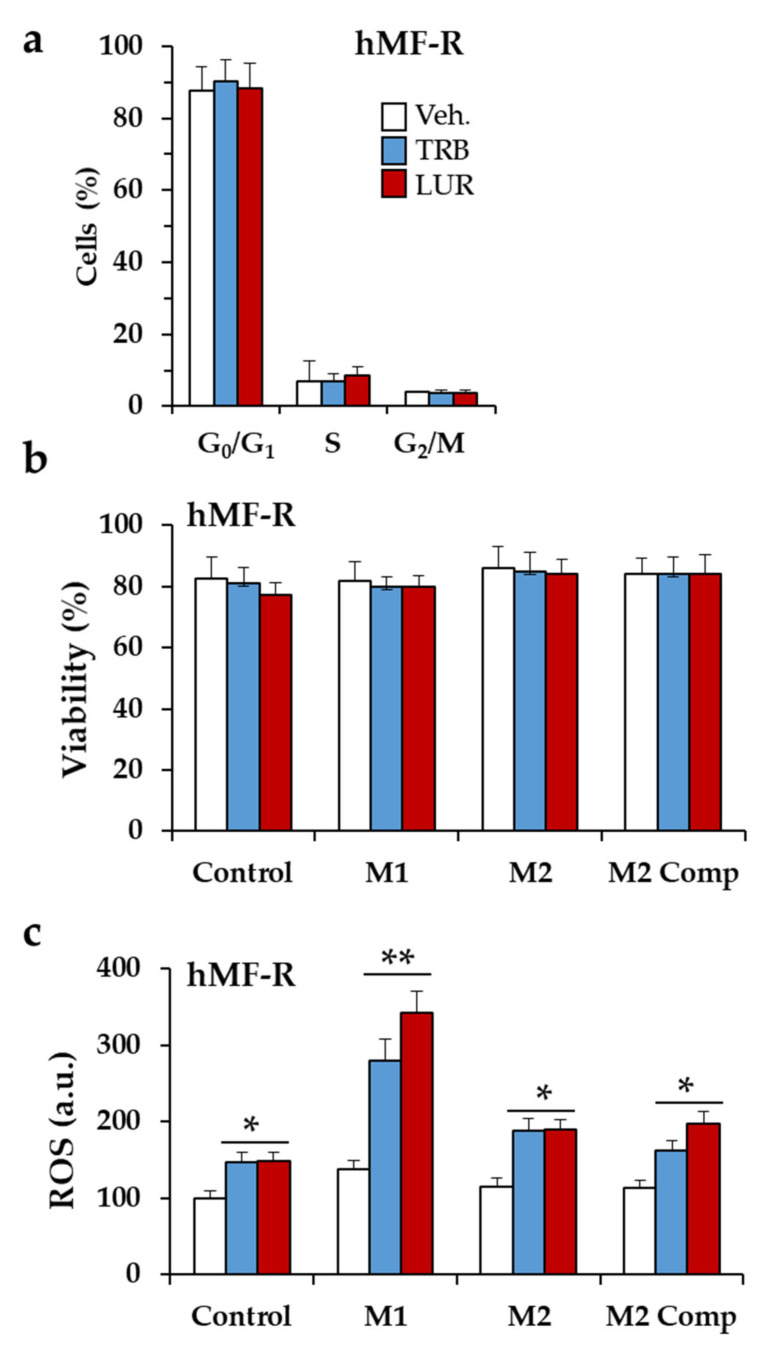
Effect of TRB and LUR on the cell cycle distribution, viability and ROS production of polarized hMF-R. (**a**) Cell cycle distribution of hMFs after 24 h treatment with TRB or LUR; (**b**) effect of TRB (5 nM) and LUR (1 nM) on viability; and (**c**) ROS production of polarized hMF-R. M1: lipopolysaccharide (LPS) (100 ng/mL) + interferon-γ (IFNγ) + interleukin-1β (IL1β) + tumor necrosis factor-α (TNFα) (20 ng/mL, each); M2: IL4 + IL13 (20 ng/mL, each); M2 Comp: M2 plus IL10 (20 ng/mL). Results show the means + S.E.M. of cells from 8 (panel a) and 12 (panels b, c) different donors. * *p* < 0.05; ** *p* < 0.01; vs. the vehicle (Veh.) condition. a.u.; arbitrary units.

**Figure 3 cancers-12-03060-f003:**
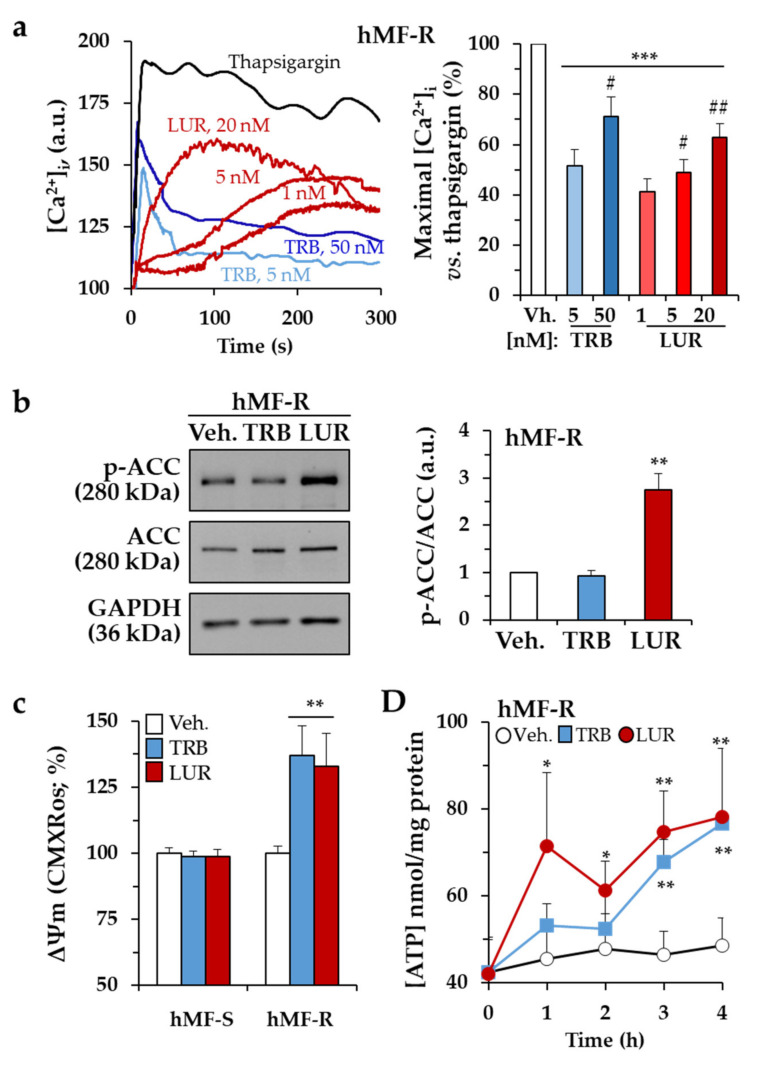
Time-lapse effects of TRB and LUR in hMF intracellular Ca^+2^ signaling and mitochondrial membrane potential. (**a**) Intracellular Ca^+2^ concentration in hMF-R after thapsigargin (200 nM), TRB, or LUR treatment; (**b**) p-ACC/ACC ratio after 30 min of treatment with TRB (50 nM) and LUR (20 nM); (**c**) hMF-R and hMF-S mitochondrial membrane potential after 24 h of treatment with TRB (5 nM) and LUR (1 nM); (**d**) intracellular ATP levels in hMF treated with TRB (5 nM) and LUR (1 nM). Results show a representative blot out of three (**b**) and the means + S.E.M. from 8 different donors, * *p* < 0.05; ** *p* < 0.01; *** *p* < 0.001 vs. the vehicle condition; ^#^
*p* < 0.05; ^##^
*p* < 0.01; vs. the corresponding condition with TRB (5 nM) or LUR (1 nM), respectively. a.u., arbitrary units.

**Figure 4 cancers-12-03060-f004:**
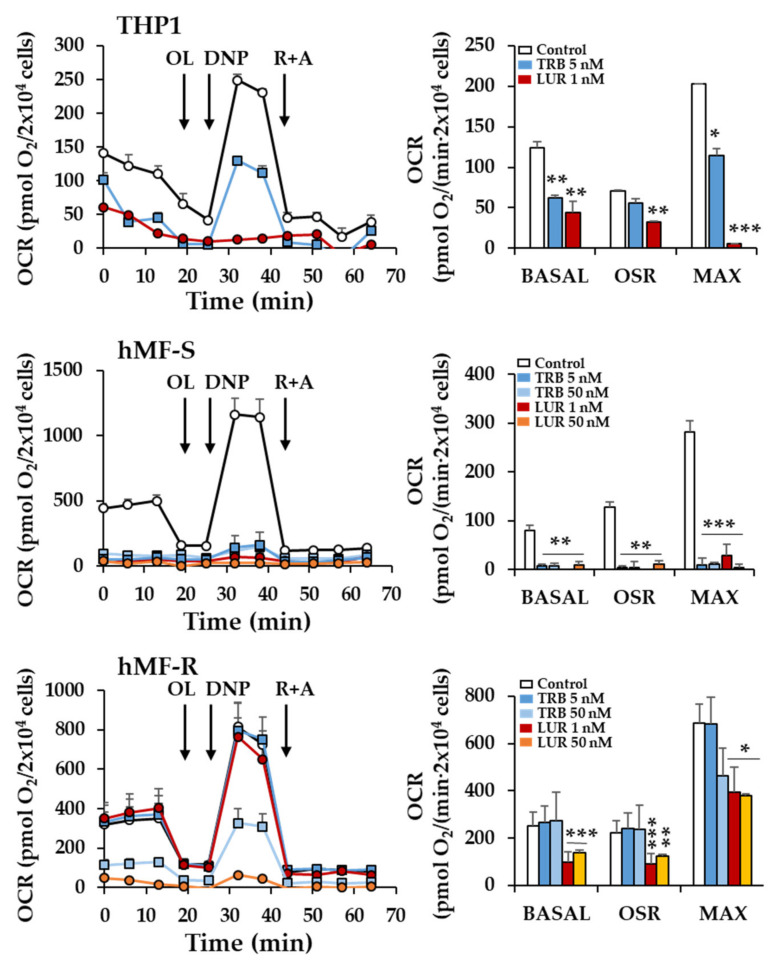
Effect of TRB and LUR in THP-1 and hMF oxygen consumption rate (OCR) after 24 h of treatment. hMF (2 × 10^4^ cells) were treated with the indicated concentrations of drugs. At the indicated times oligomycin (OL), 2′,4′-dinitrophenol (DNP), and rotenone plus antimycin (R+A) were added. (Right panels) basal respiration (BASAL); OSR (oligomycin sensitive respiration); MAX (maximal respiration). Results show the means + S.E.M. from three THP-1-independent experiments by triplicate and from 7 different healthy donors by triplicate. * *p* < 0.05; ** *p* < 0.01; *** *p* < 0.001 vs. the vehicle condition.

**Figure 5 cancers-12-03060-f005:**
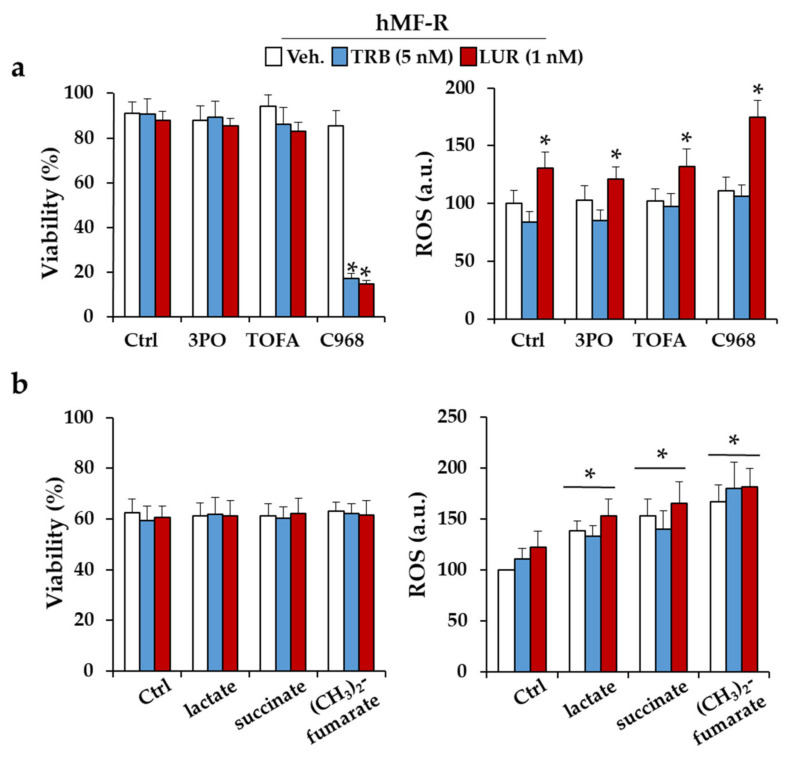
Immuno-metabolic effects of TRB and LUR on hMF inhibition of glycolysis, fatty acid biosynthesis, and glutamine metabolism. (**a**) hMFs were treated with the indicated inhibitors, and viability and ROS production were measured at 24 h; 10 µM 3PO (3-pyridin-3-il-1-piridin-4-ilprop-2-en-1-one; a PFKFB-3 inhibitor); 1 µg/mL TOFA (5-tetradeciloxi-2-furoicacid; an ACC inhibitor); 40 µM C968 (968 compound; a mitochondrial glutaminase inhibitor); (**b**) immuno-metabolic modulation assay with lactate (20 µM), succinate (100 µM), and dimethyl-fumarate (100 µM). hMF viability and ROS production were measured at 24 h. Results show the means + S.E.M. from 5 different donors, * *p* < 0.05 vs. the control immune-metabolic untreated condition. a.u., arbitrary units.

**Figure 6 cancers-12-03060-f006:**
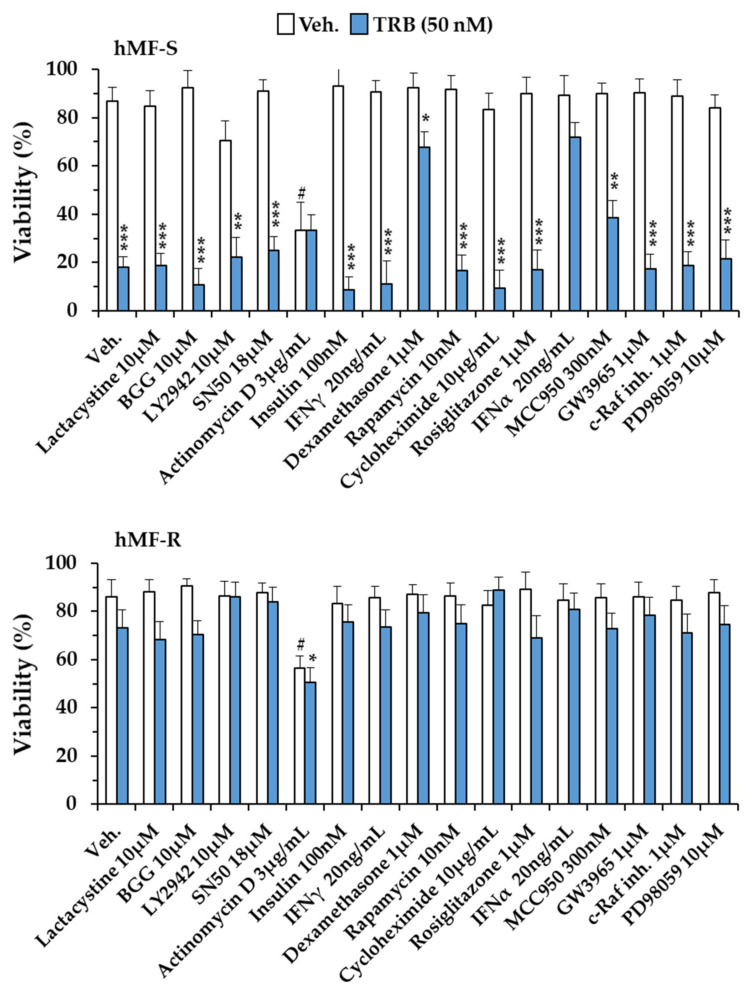
Analysis of different pathway targeting on the viability of hMFs treated with TRB for 24 h. hMFs from either sensitive (hMF-S; *n* = 5) or resistant (hMF-R; *n* = 12) donors were treated with the indicated compounds, and cell viability was determined by flow cytometry. Results show the means + S.E.M. from the indicated number (*n*) of healthy donors for each group of hMFs. * *p* < 0.05; ** *p* < 0.01; *** *p* < 0.001 vs. the corresponding to the vehicle condition; ^#^
*p* < 0.05 vs. the Veh. condition (drug-untreated condition).

**Figure 7 cancers-12-03060-f007:**
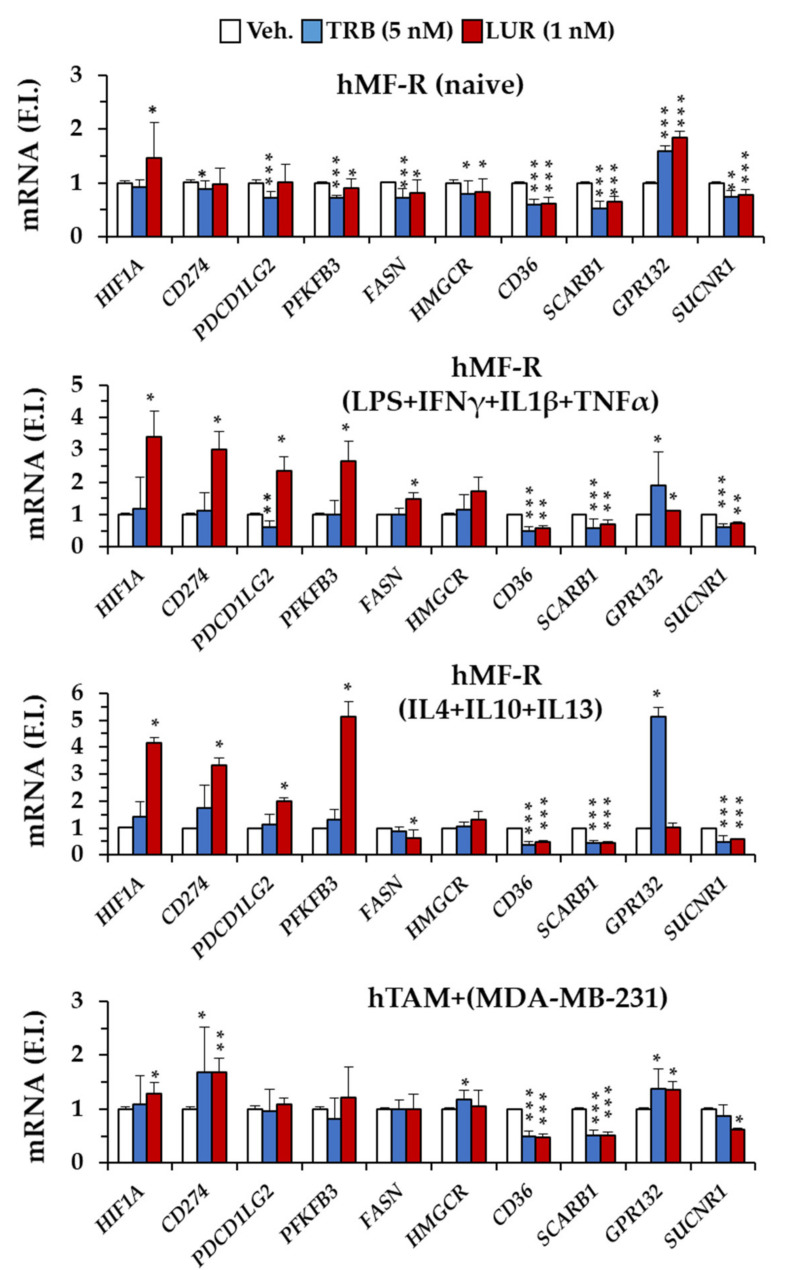
hMF differential gene expression patterns in naïve and polarized cells after 24 h of treatment with TRB and LUR. hMF-RNAs from healthy donors were isolated and underwent RT. cDNA were obtained and 3 pools of the same amount of cDNA were prepared. Each cDNA pool consisted of 4 donors. Treatments: M1: LPS (100 ng/mL) + IFNγ + IL1β + TNFα (20 ng/mL, each); M2: IL4 + IL-10 + IL13 (20 ng/mL, each); hTAM: hMF + MDA supernatants 24 h prior to the experiment. Results show the means + S.E.M. of fold induction (F.I.) from 8 to 12 different healthy donors assayed per triplicate. * *p* < 0.05; ** *p* < 0.01; *** *p* < 0.001 vs. the vehicle (Veh.) condition.

**Figure 8 cancers-12-03060-f008:**
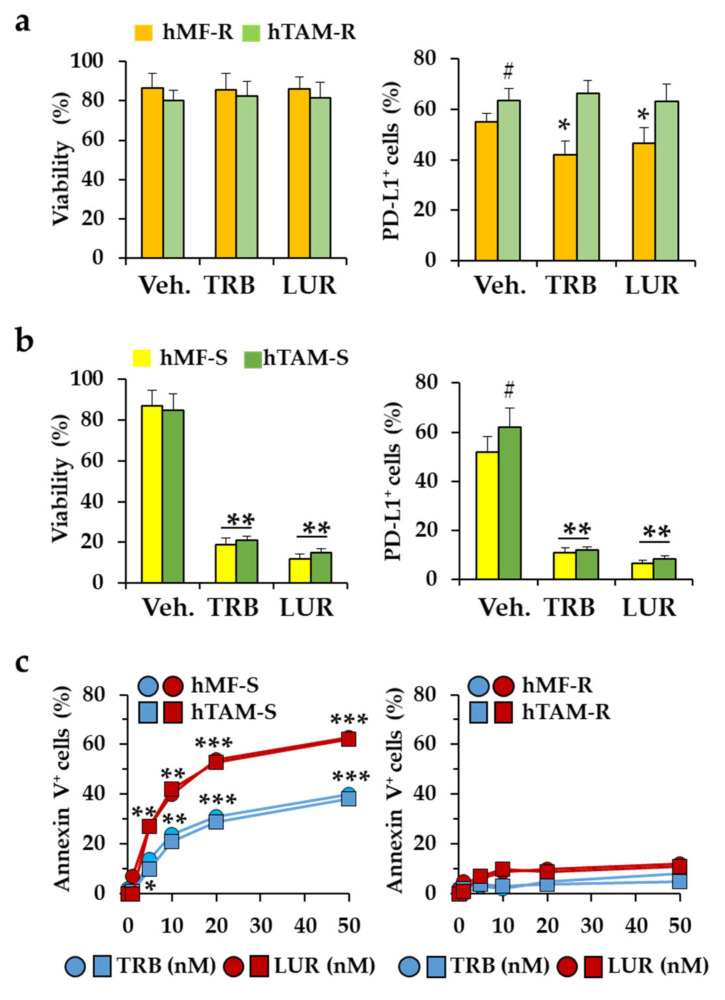
Effect of TRB and LUR on viability and cell surface levels of programmed-death ligand 1 (PD-L1) in hMF. (**a**) hMF and hTAM differentiated cells from drug-resistant donors were treated for 24 h with TRB (5 nM) and LUR (1 nM) and the viability and PD-L1 exposure in the cell surface were determined by flow cytometry; (**b**) analysis in TRB and LUR sensitive (hMF-S) cells; (**c**) dose-dependent effect of TRB and LUR on annexin V exposure of hMFs and hTAMs from hMF-R and hMF-S healthy donors. Results show the means + S.E.M. of 8 independent preparations of cells. * *p* < 0.05; ** *p* < 0.01; *** *p* < 0.005 vs. the vehicle (Veh.) condition; ^#^
*p* < 0.05 vs. the hMF corresponding condition.

**Figure 9 cancers-12-03060-f009:**
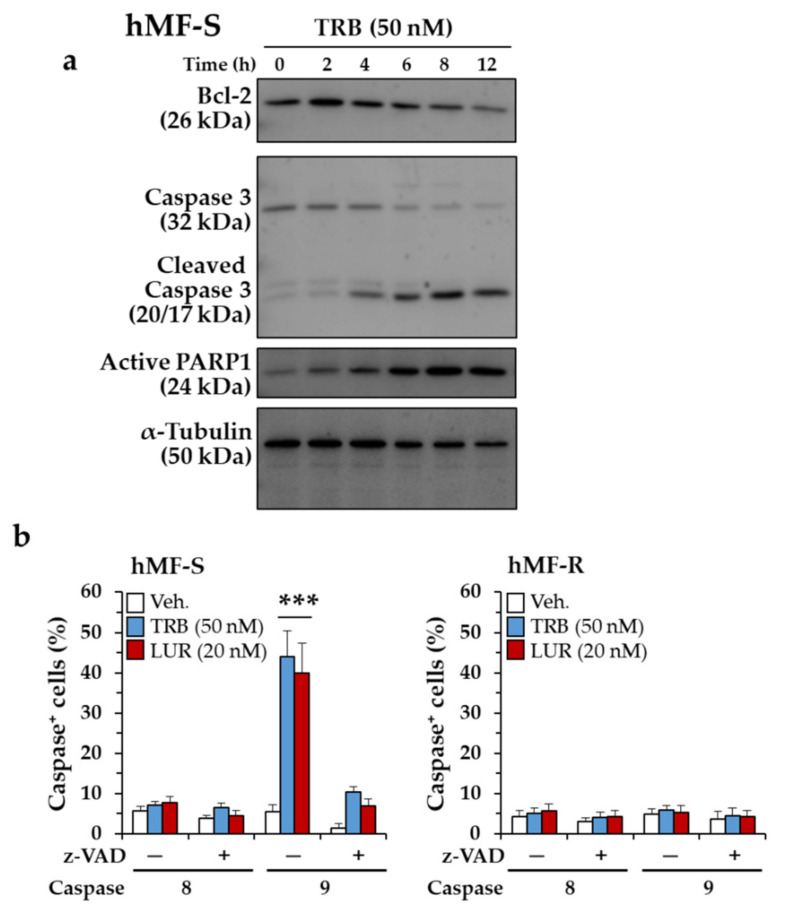
TRB and LUR administration promotes apoptosis in hMF-S cells. (**a**) hMF-S cells were treated with TBR and the protein levels of B-cell lymphoma 2 (Bcl-2), the cleavage of caspase 3, and the accumulation of the poly (ADP)-ribose polymerase 1 (PARP1) cleaved protein were determined by immunoblot at the indicated times; (**b**) hMF-S and hMF-R cells were treated with TRB (50 nM) and LUR (20 nM) and loaded with permeant caspase 8 and caspase 9 activity-sensitive fluorescence substrates, and their activity was determined by flow cytometry at 8 h. Results show the means + S.E.M. of 5 independent preparations of each type of cells. *** *p* < 0.005 vs. the vehicle (Veh.) condition.

**Table 1 cancers-12-03060-t001:** Primary and secondary antibodies (supplier, reference, and dilution used).

Antibody; Provider; Reference (Dilution)
anti-Bcl-2; Cell Signaling (Rabbit mAb); ref. 2870 (1:1000)
anti-caspase 3; Cell Signaling (Rabbit mAb); ref. 9662 (1:1000)
anti-cleaved caspase 3; Cell Signaling (Rabbit mAb); ref. 9661 (1:1000)
anti-PARP-1; R&D Systems (Goat pAb); ref. AF-600-NA (1:1000) anti-PACC; Cell Signaling (Rabbit mAb); ref. 3661 (1:1000)
anti-ACC; Cell Signaling (Rabbit mAb); ref. 3676 (1:1000)
anti-RNA polymerase II; Abcam; ref. ab5095 (1:1000)
anti-α-tubulin; Cell Signaling (Rabbit mAb); ref. D7F10 (1:500) anti-GAPDH; Invitrogen (Mouse mAb); ref. 39-8600 (1:1000)
anti-rabbit IgG; Sigma (pAb); ref. A6154 (1:5000) anti-mouse IgG; Invitrogen (Mouse mAb); ref. A9044 (1:5000)

**Table 2 cancers-12-03060-t002:** mRNA primer sequences (human sequences; 5′–3′).

mRNA	Forward Primers Sequences	Reverse Primers Sequences
*CD274*	TGGCATTTGCTGAACGCATTT	TGCAGCCAGGTCTAATTGTTTT
*PD-CD1LG2*	ATTGCAGCTTCACCAGATAGC	AAAGTTGCATTCCAGGGTCAC
*TLR-4*	TTTGGACAGTTTCCCACATTGA	AAGCATTCCCACCTTTGTTGG
*IL10*	CGAGATGCCTTCAGCAGAGT	CGCCTTGATGTCTGGGTCTT
*SUCNR1*	GGAGACGTGCTCTGCATAAG	AGGTGTTCTCGGAAAGGATACTT
*HIF1A*	GAACGTCGAAAAGAAAAGTCTCG	CCTTATCAAGATGCGAACTCACA
*FASN* *CD36*	AAGGACCTGTCTAGGTTTGATGC CTTTGGCTTAATGAGACTGGGAC	TGGCTTCATAGGTGACTTCCA GCAACAAACATCACCACACCA
*HMGCR*	TGATTGACCTTTCCAGAGCAAG	CTAAAATTGCCATTCCACGAGC
*LDLR*	TCTGCAACATGGCTAGAGACT	TCCAAGCATTCGTTGGTCCC
*PFKFB3*	TTGGCGTCCCCACAAAAGT	AGTTGTAGGAGCTGTACTGCTT
*SCARB1*	AATAAGCCCATGACCCTGAAGC	GCCCCACATGATCTCACCC
*TNFA*	CCAGAGGGAAGAGTTCCCCAGGG	AGGCTTGTCACTCGGGGTTCGAG
*GPR132*	TGTTCCAGACGGAAGACAAGG	GCGTAGTAGTACCCGGCAA
*RPLP0*	CAGGCGTCCTCGTGGAAGTGAC	CCAGGTCGCCCTGTCTTCCCT

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
