# Peer review of "Specific Effects of Trabectedin and Lurbinectedin on Human Macrophage Function and Fate—Novel Insights"

_cancers, 2020, doi:10.3390/cancers12103060_

Round 1
Reviewer 1 Report
In this study, the authors investigate the effects of two chemotherapy drugs, Trabectedin and Lurbinectedin, on the monocyte-derived macrophages derived from human patients. Their suggest that the actions of these drugs on these cell populations could potentially yield insights into why some patients experience severe neutropenia or monocytopenia when taking these drugs. The authors obtain monocyte-derived macrophages from 160+ human donors, treat these macrophages with Trabectedin or Lurbinectedin and assay their effects using a variety of functional assays.
The authors find that macrophages from cells different donors showed different susceptibilities to drug treatment, with some being sensitive and others being resistant to drug treatment. They then find effects of these drugs on calcium dynamics, mitochondrial function metabolism, cell death and gene expression.
A fundamental limitation of this study is its assumption that macrophages are the main targets of these drugs in vivo. This is not necessarily true, as it is possible that selective effects on hematopoietic stem cells, not mature cells, might likely be the causes of neutropenia or monocytopenia. If the authors have any reason to believe that these drugs are targeting tumor-associated macrophages, they should introduce more fully.
Notwithstanding, it was interesting that cells from different donors showed differential susceptibility to death as a result of drug treatment; however, the presented results did not adequately convey the effects and degree of heterogeneity across different individuals. I had these major concerns:
- Specifically, I found it unclear how patient cells grouped into sensitive and resistant categories. It would be more informative if the cell viability of each individual population could be reported directly, and with representative flow plots, and the point of cut off for grouping cells into these two categories clearly delineated.
- The authors used a 11-14 day in vitro culture protocol to derive macrophages from monocytes; however, from reading the paper and methods, I found it unclear how this protocol would result in macrophages, as adherent peripheral blood mononuclear cells were simply cultured for 14 days in cell culture medium. If this protocol has been validated elsewhere, it needs to be cited.
- In addition, it would be important to see an end-point flow cytometry analysis to confirm that macrophages were obtained using this protocol.
- From Figures 2 onwards, it was sometimes unclear whether experiments were performed on sensitive or resistant populations (or both). This needs to be specified in each bar chart. In addition, for each bar chart, it needs to be clear how many human cell samples were analyzed (e.g. N=??)
Author Response
Please see the file Comments to Reviewer 1

Reviewer 2 Report
The manuscript "Selective apoptotic targeting of human macrophages by trabectedin and lurbinectedin – relevance for the therapeutic action of these drugs" by Povo-Retana, A. et al. deals with the differential response of primary human macrophages to the therapeutically relevant drugs trabectedin and lurbinectedin. The authors have used scientifically sound methodology to understand macrophage function and polarization. This topic is of great interest to readers and might significantly contribute to the current knowledge in the field.
I have a few significant concerns with this work that are outlined below:
- Based on the huge amount of data presented in the manuscript regarding macrophage response to the drugs, I do not understand the claim made in the title. I guess that a more broadly selected title would better fit to the data presented in the manuscript.
- The introduction is not very informative: please provide more detailed information and references on the role of macrophages for cancer. There is nearly nothing mentioned, and if so, it is very broad. There is already a lot of literature regarding trabectidin and macrophages in cancer as well as macrophages as therapeutic targets in the tumor context. Please stick more to the context of cancer than broadly saying that macrophages have PRRs and recognize PAMPs and are involved in inflammatory responses. This is not very helpful for the understanding of the main message of the manuscript. Please provide also more information about lubrinectedin and its mode of action.
- The M1/M2 classification should be avoided. This is very old-fashioned and far too simple for macrophage biology. I would prefer pro- versus anti-inflammatory macrophages or you could stick to the way you presented in figure 7, stating simply the way how macrophages were stimulated.
- Figure 1a: I do not see the “moderate” rise in ROS. I can only appreciate not regulated or reduced levels of ROS. Please revise and explain. Figure 1c: why did the authors use the mentioned concentrations of TRB 100nM and LUR 50nM. Please explain. In general: Could you also explain the rather different response of macrophages to LUR versus TRB? Has this something to do with the mode of action of the two drugs?
- Please be precise regarding sample numbers. Do not state “among more than 164 individuals” or “nearly 200” if it’s only 164. The same is true for n-numbers. Do not use 8-12: what does this mean? Did you exclude values for some of the experiments? Please go through the text and change this accordingly.
- Figure 2: Why did the authors change the concentrations of drugs used for this figure and all following figures to TRB 5nM and LUR 1nM? Please explain, also in the manuscript.
- Figure 6: This is very huge figure with lots of graphs and significances. It would help the reader if you could draw a line through the figure from the only TRB-treated cells are so that co-Stimulation with other agents can be appreciated much easier. Also, put in significances only for TRB alone versus TRB+agent (e.g. dexamethasone). Also the effects of actinomycin D on both vehicle and TRB treatment should be explained in the results section. Another point is: Why are there only TRB-treated cells and not LUR-treated cells? Could you please provide these data as well?
- Figure 7: Again, it would be helpful to better understand the results, if the authors draw a line through the figure where the vehicle baseline is. I would also prefer to have the y-axes all the same sizes so that comparison between pre-treatments is more easily achievable. Maybe you also consider to segment the y-axis since minor changes are only poorly visible. The same is true for errors bars. Some seem to have, some not. Maybe this I because of the same reason.
- Did the authors had a chance to look into primary TAM? Maybe from a more physiological model such as mouse model or at least 3D in vitro setting? Or maybe in already existing publically available data sets? If this kind of data could be added, it would really strengthen the content of the paper.
- Discussion: I would appreciate if the authors could include more speculation about what this really means for primary TAM? And how this knowledge might be applicable to the clinical setting. Also the discussion about PD-L1 and the role of checkpoints in immunotherapy is only poorly addressed.
- Material and Methods: p.15 l. 378 TNF-alpha is mentioned. However, in the manuscript, there is no experiment with TNF-alpha. Please correct.
- Material and Methods: in polarization experiments – have the macrophages been washed before TRB or LUR treatment? This issue might have effects on the results if cytokines remained longer or were present upon treatment with TRB or LUR and needs to be included.
Author Response
Please see the file Comments to Reviewer 2

Round 2
Reviewer 2 Report
The authors fully addressed my previous concerns and the manuscript can now be accepted for publication in its present form.